# Bone morphogenetic protein 1.3 inhibition decreases scar formation and supports cardiomyocyte survival after myocardial infarction

Slobodan Vukicevic[1,10], Andrea Colliva[2,3,10], Vera Kufner [1], Valentina Martinelli[4], Silvia Moimas [4], Simone Vodret[2], Viktorija Rumenovic [1], Milan Milosevic [5], Boris Brkljacic [6], Diana Delic-Brkljacic[7], Ricardo Correa [2], Mauro Giacca[3,4,8], Manuel Maglione[9], Tatjana Bordukalo-Niksic [1], Ivo Dumic-Cule [1,11] & Serena Zacchigna [2,3,11✉]

Despite the high prevalence of ischemic heart diseases worldwide, no antibody-based treatment currently exists. Starting from the evidence that a specific isoform of the Bone Morphogenetic Protein 1 (BMP1.3) is particularly elevated in both patients and animal models of myocardial infarction, here we assess whether its inhibition by a specific monoclonal antibody reduces cardiac fibrosis. We find that this treatment reduces collagen deposition and cross-linking, paralleled by enhanced cardiomyocyte survival, both in vivo and in primary cultures of cardiac cells. Mechanistically, we show that the anti-BMP1.3 monoclonal antibody inhibits Transforming Growth Factor β pathway, thus reducing myofibroblast activation and inducing cardioprotection through BMP5. Collectively, these data support the therapeutic use of anti-BMP1.3 antibodies to prevent cardiomyocyte apoptosis, reduce collagen deposition and preserve cardiac function after ischemia.

[1] Laboratory for Mineralized Tissues, Center for Translational and Clinical Research, University of Zagreb School of Medicine, Zagreb, Croatia. [2] Cardiovascular Biology, International Centre for Genetic Engineering and Biotechnology, Trieste, Italy. [3] Department of Medicine, Surgery and Health Sciences, University of Trieste, Trieste, Italy. [4] Molecular Medicine, International Centre for Genetic Engineering and Biotechnology, Trieste, Italy. [5] Department for Environmental Health and Occupational and Sports Medicine, University of Zagreb School of Medicine, Zagreb, Croatia. [6] Department of Diagnostic and Interventional Radiology, University Hospital Dubrava, University of Zagreb School of Medicine, Zagreb, Croatia. [7] Department of Cardiology, Clinical Hospital Sisters of Mercy, University of Zagreb School of Medicine, Zagreb, Croatia. [8] School of Cardiovascular Medicine & Sciences, King's College London, London, UK. [9] Department of Visceral, Transplant and Thoracic Surgery, Center of Operative Medicine, Medical University of Innsbruck, Innsbruck, Austria. [10] These authors contributed equally: Slobodan Vukicevic and Andrea Colliva. [11] These authors contributed equally: Ivo Dumic-Cule and Serena Zacchigna. ✉ email: zacchign@icgeb.org

D espite substantial progress in the prevention of cardiovascular diseases during the last few decades, 50 million patients experience acute myocardial infarction (MI) worldwide every year (http://www.who.int/cardiovascular_diseases/en/). The lost myocardium is replaced by a fibrotic scar, which further expands, leading to heart dysfunction and life-threatening complications. Therapeutic options to prevent, slow down and reverse disease progression following MI have remained limited. Progressive loss of contractile function following cardiac injury is a consequence of the poor regenerative capacity of cardiomyocytes and their replacement by a collagen-rich fibrotic tissue. Multiple attempts have been made to regenerate the heart using various strategies, which include implantation of exogenous cells[1], induction of hypoxia[2], stimulation of pro-regenerative genetic programs[3–5], injection of extracellular matrix (ECM) components[6], and administration of growth factors[7,8]. However, none of these approaches has been successfully translated into human therapy yet. On the other hand, very few therapeutic strategies have focused on fibrosis modulation. Despite the existence of over 500 ongoing cardiovascular clinical trials (http://clinicaltrials.gov), there are no effective and safe drugs to treat fibrosis[9–11]. Monoclonal antibodies represent the most successful example of biological drugs, which are offering hope for the treatment of various complex disorders, including rheumatological diseases and cancer, and they feed the fastest growing sector in the pharmaceutical industry.

Fibrotic remodeling after MI starts with dead cardiomyocytes being cleared by macrophages and progressively replaced by reparative cells, largely fibroblasts, that are activated into myofibroblasts, which acquire a contractile phenotype through the expression of α-smooth muscle actin (αSMA) and start secreting abundant ECM proteins[12]. Scar formation is critical to protect the heart from life-threatening complications, such as aneurysm formation and cardiac rupture, yet it boosts progressive adverse remodeling, often resulting in ventricle dilation and failure.

Members of the transforming growth factor β (TGFβ) superfamily play an essential role in fibrotic processes[13]. Although TGFβ1 is the best-studied factor after MI, emerging evidence points to a major involvement of other family members, including various bone morphogenetic proteins (BMPs), in cardioprotection and post-infarction remodeling[14].

As reflected by their name, BMPs possess well characterized osteogenic properties[15]. Yet, genetic loss-of-function models have revealed their essential role in cardiovascular development[16,17]. In addition, multiple evidence indicates that various BMPs, as well as other TGFβ superfamily members, are modulated after MI[18,19] and control cardiomyocyte survival[20–22].

Different from other BMPs, BMP1 does not belong to the TGFβ superfamily, as it does not share significant amino acid sequence homology with other family members [23]. Instead, BMP1 is a pro-collagen C-proteinase, acting as a zinc metalloproteinase that cleaves the carboxyl pro-domains of procollagens to produce mature monomers of the major fibrillar collagens[24,25]. This cleavage is essential for the proper assembly of insoluble collagen within the ECM and scar formation. More recently, multiple splicing variants of the same gene have been identified and named with sequential suffixes, from BMP1.1 (originally discovered in bone) to BMP1.7[24]. Both BMP1.1 and its longer isoform BMP1.3 convert a variety of ECM precursors into mature functional proteins, including pro-collagen C I–III, small leucine-rich proteoglycans (decorin, biglycan), laminin, collagen VII and perlecan in the basal membrane, the BMP antagonist chordin, and pro-lysyl oxidases, which mediate collagen crosslinking[26]. Both isoforms were detected as soluble proteins in the blood of healthy individuals, as well as of patients[27]. A general, yet unproven, hypothesis assumes that each BMP1 isoform might modulate fibrotic events in different organs[28]. For instance, antagonism of circulating BMP1.3 reduced fibrosis in animal models of chronic kidney disease[27] and liver cirrhosis[29]. The involvement of various BMP1 isoforms in repair, remodeling and fibrosis of the infarcted heart remains poorly investigated.

By liquid chromatography–mass spectrometry we detected the presence of BMP1.3 in human plasma[30] and, more recently, developed an ELISA assay to quantify the circulating levels of this isoform. Using this assay, we found increased levels of BMP1.3 in the plasma of patients with MI, suggesting that this factor could modulate cardiac fibrosis. This provided the rationale for developing and testing the therapeutic efficacy of an anti-BMP1.3 monoclonal antibody in the modulation of fibrosis following MI. Here we show that this anti-BMP1.3 antibody prevents cardiomyocyte apoptosis, reduces collagen deposition, and preserves cardiac function after ischemia.

## Results

**BMP1.3 levels are increased in both patients and mice after MI.** To explore the involvement of BMP1.3 in cardiac fibrosis, we exploited our in house-developed ELISA (validation in Supplementary methods) to quantify BMP1.3 levels in the plasma of either healthy individuals or patients who experienced a recent MI, as diagnosed by ST-elevation at electrocardiography and increased troponin T concentration. Blood sampling was performed in patients referring to emergency unit, within 12 h after hospitalization for MI. Significantly increased BMP1.3 levels were detected in patients with MI compared to healthy individuals (Fig. 1a).

To verify the source of circulating BMP1.3, we extracted RNA from post-mortem heart samples of patients either affected by ischemic cardiomyopathy or not affected by any overt cardiac disease. As shown in Fig. 1b, *BMP1.3* mRNA levels were higher in ischemic than in non-ischemic hearts.

Similar findings were observed in mice subjected to MI, by surgically ligating the left anterior descendent (LAD) coronary artery. Also in this model, BMP1.3 levels appeared increased after cardiac ischemia in both circulating blood and heart (Fig. 1c, d). To define the cellular source of BMP1.3, we isolated cardiomyocytes, endothelial cells, fibroblasts, and inflammatory cells from hearts at 2 days after MI. As shown in Fig. 1e, fibroblasts produced by far the highest levels of *Bmp1.3*.

**Anti-BMP1.3 antibody reduces fibrosis and preserves cardiac function in rodent models of MI.** To assess whether the inhibition of BMP1.3 can be exploited to modulate the fibrotic response after cardiac injury, we produced a mouse monoclonal antibody using the BMP1.3 peptide, corresponding to amino acids 972-986 of the full-length protein, as immunogen, following standard hybridoma technique[31]. The neutralizing properties of the antibody were demonstrated by a neutralization assay, in which the capacity of BMP1.3 to cleave its substrate dentin matrix protein-1 (DMP-1) was completely inhibited by the anti-BMP1.3 antibody (Supplementary Fig. 1a, b). Pharmacokinetic analysis in rats showed that the antibody can be efficiently delivered by intravenous injection, reaching a peak 3–6 h after administration and persisting at a lower concentration up to 3 days (Supplementary Fig. 1c). Intraperitoneal injection resulted in more variable and lower blood concentrations (Supplementary Fig. 1d). Thus, in the next experiments, we administered the antibody intravenously according to the timeline shown in Supplementary Fig. 2a.

The therapeutic effect of the antibody was tested in two rodent models of cardiac damage. First, we administered two consecutive doses of isoproterenol, known to exert massive chronotropic and inotropic activity, thus inducing extensive myocardial necrosis and fibrosis, in rats[32–34]. Anti-BMP1.3 antibody was injected as

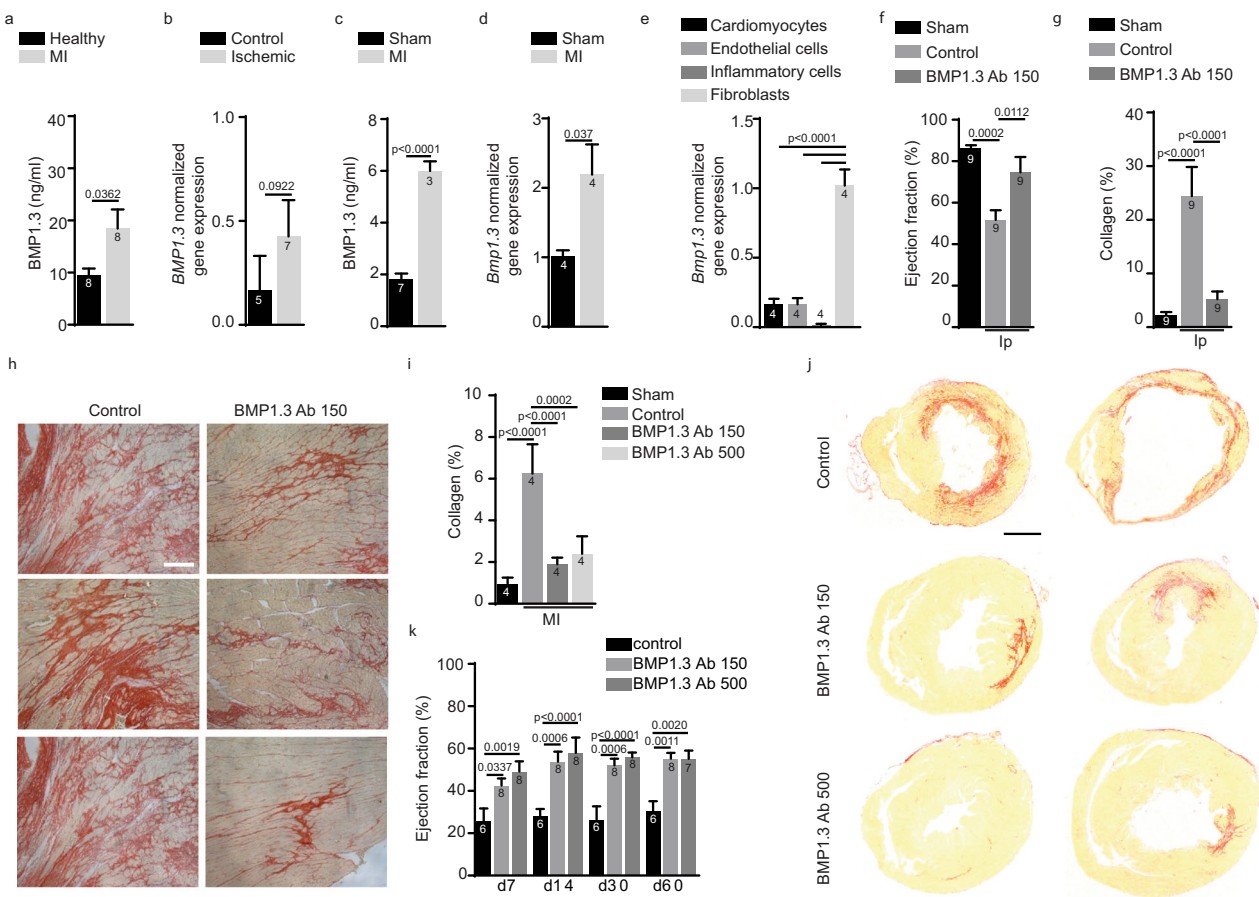

**Fig. 1 Therapeutic activity of anti-BMP1.3 antibody after cardiac damage in vivo. a** Quantification of BMP1.3 protein levels in blood of healthy individuals and patients affected by acute myocardial infarction (MI) within 12 h from hospitalization. **b** Quantification of the expression levels of *BMP1.3* normalized over *GAPDH* in human myocardium of patients without overt cardiac disease (control) or affected by ischemic heart disease (ischemic). **c** Quantification of BMP1-3 protein levels in the blood of sham-operated or infarcted mice at 24 h after MI. **d** Quantification of the expression levels of *Bmp1.3* normalized over *18 S* in the myocardium of sham-operated or infarcted mice at 48 h after MI. **e** Quantification of the levels of *Bmp1.3* mRNA normalized on *Actb* in different cardiac cell types at 48 h after MI. **f** Quantification of the ejection fraction in rats treated with isoproterenol (Ip), alone (control) or in combination with anti-BMP1.3 antibody (150 µg/kg). Values for sham animals are also shown for reference. **g** Quantification of the area covered by collagen in each section. **h** Representative images of heart sections stained with Sirius red to show collagen deposition upon administration of isoproterenol alone (control) or in combination with anti-BMP1.3 antibody. **i** Quantification of the area covered by collagen in sections of murine hearts sham-operated or injured by myocardial infarction (MI) and treated with the indicated dose of anti-BMP1.3 antibody (150 or 500 µg/kg). **j** Representative images of murine heart sections stained with Sirius red to show collagen deposition upon MI and treatment with the indicated dose of anti-BMP1.3 antibody. **k** Quantification of the ejection fraction at 7, 14, 30, and 60 days after MI. Data in **a**, **b**, **c**, **d**, **e**, **f**, **g**, **i**, and **k** are shown as mean ± s.e.m. Sample number is indicated inside or above each bar. Statistical significance was determined using unpaired, two-sided *t*-test in **a–d**, one-way ANOVA, followed by Tukey's test in **e**, **f**, **g**, **i**, and two-way ANOVA in **k** followed by Tukey's multiple comparison test. Scale bar in **h**, **j** is 200 µm and 1 mm, respectively. Source data are provided as a Source Data file.

described in Supplementary Fig. 2a, starting at 2 h after isoproterenol administration. Isoproterenol injection resulted in pronounced increase in circulating troponin T levels, with a peak at day 1 and persisting up to day 3 after the last administration (Supplementary Fig. 2b). Rats that received the anti-BMP1.3 antibody showed significantly lower troponin T levels at both 24 and 48 h (Supplementary Fig. 2b) and preserved ejection fraction, as measured by echocardiography at 1 month (Fig. 1f). Quantification of cardiac fibrosis in sections stained with Sirius red indicated that the collagen area was markedly reduced in anti-BMP1.3 treated hearts compared to control animals (Fig. 1g, h). Treatment with anti-BMP1.3 antibody reverted both parameters, ejection fraction and fibrosis, back to values similar to those observed in sham-operated animals (Fig. 1f, g).

Next, we validated the therapeutic activity of the anti-BMP1.3 antibody after MI in mice. The anti-BMP1.3 antibody was repeatedly administered at either low (150 µg/kg) or high (500 µg/

kg) dose, according to the protocol shown in Supplementary Fig. 2a. Morphometric and histological analysis at 2 months after MI showed a significant deposition of fibrotic tissue in the left ventricle (LV) of control animals, as expected (Fig. 1i, j). Treatment with any dose of anti-BMP1.3 antibody significantly reduced scar size (Fig. 1i, j), in line with data obtained in the rat model. Cardiac function was monitored by echocardiography up to 2 months. The main parameters of cardiac function, ejection fraction and fractional shortening, were significantly higher in animals treated with either dose of the anti-BMP1.3 antibody compared to control animals at all time points (Fig. 1k and Supplementary Fig. 2c). Accordingly, additional indexes of cardiac function, including LV end-systolic and end-diastolic diameters (LVIDs and LVIDd), LV systolic and diastolic anterior wall thickness (LVAWs and LVAWd) and LV anterior wall thickening were all improved in mice treated with the anti-BMP1.3 antibody (Supplementary Fig. 2d–h).

We also evaluated how anti-BMP1.3 antibody treatment performed in comparison with alternative approaches blocking TGFβ, the major trigger of fibrosis. We performed MI in a new set of 16 mice and randomly assigned them to receive the following treatments: (i) anti-BMP1.3 antibody; (ii) anti-TGFβ1 antibody; (iii) SB-431542, a chemical inhibitor specifically blocking the TGFβ receptors Alk4, Alk5, and Alk7, (iv) IgG1 antibody as isotype control. As shown in Supplementary Fig. 3, anti-BMP1.3 treatment was the only approach that significantly improved cardiac function at 2 weeks after MI.

**Anti-BMP1.3 antibody inhibits myofibroblast activation via downregulation of TGFβ pathway**. Because collagen I is the major component of cardiac scar, we assessed the effect of the anti-BMP1.3 antibody on its expression, taking advantage of a reporter mouse, in which the enhanced green fluorescence protein (EGFP) transgene is expressed under the control of the collagen α1(I) promoter (COLL-EGFP mice). The administration of the anti-BMP1.3 antibody in these mice significantly reduced collagen expression after MI, as shown by the levels of EGFP fluorescence in heart sections (Fig. 2a, b).

We then crossed these mice with another reporter strain, in which expression of red fluorescent protein (RFP) is driven by the αSMA promoter. In these dual reporter mice, myofibroblast activation can be monitored and quantified as increase in both EGFP and RFP fluorescence, indicative of collagen I and αSMA overexpression, respectively. Primary cardiac fibroblasts from dual reporter mice were kept in culture and activated by the administration of TGFβ, as described[35]. The addition of recombinant BMP1.3 protein to these cells further increased both collagen and αSMA expression. In contrast, exposure to the anti-BMP1.3 antibody resulted in a significant downregulation of both collagen and αSMA (Fig. 2c, d).

Consistent with the prominent role of TGFβ in myofibroblast activation, we assessed whether the anti-BMP1.3 antibody interfered with TGFβ pathway. By real-time PCR we detected a significant reduction in the expression of multiple pro-fibrotic genes, including *collagen-α1 (Col1a), lysyl oxidase (Lox), connective tissue growth factor (Ctgf), fibronectin (Fn)* and *Tgfβ1* itself in primary fibroblasts exposed to the anti-BMP1.3 antibody (Fig. 2e). A consistent effect was observed in vivo, as hearts subjected to MI and treated with the anti-BMP1.3 antibody showed reduced levels of Tgfβ expression, reduced phospho-Smad2 accumulation in fibroblast nuclei within the scar, and reduced expression of the TGFβ target genes *Col1a, Lox, Ctgf* and *Fn* (Fig. 2f, g). We also confirmed reduced phosphorylation of Smad2/3 in murine fibroblasts exposed to the anti-BMP1.3 antibody by Western Blotting (Supplementary Fig. 4a) and used a luciferase assay for Smad2/3 activation to show that recombinant BMP1.3 activated, while anti-BMP1.3 antibody inhibited, Tgfβ pathway in HEK293T reporter cells (Supplementary Fig. 4b).

Since Lox is the major enzyme responsible for collagen crosslinking in the scar, we thought that its downregulation could lead to a less mature and more elastic scar, consistent with the observed improvement in cardiac function. We first confirmed reduced Lox expression by immunohistochemistry in the scar of mice treated with anti-BMP1.3 antibody compared to control animals (Fig. 2h). Next, we dissected scar tissue from the infarcted heart of control and anti-BMP1.3 antibody-treated animals and analyzed collagen crosslinking. All crosslinking indicators (dihydroxynorleucine-DHNL, dihydroxylysinonorleucine-DHLNL, hydroxylysinonorleucine-HLNL, pyridinoline-Pyr, deoxypyridinoline-d-Pyr) and total aldehyde were significantly reduced by anti-BMP1.3 antibody treatment (Fig. 2i).

Thus, inhibition of BMP1.3 prevents myofibroblast activation, decreases the expression of pro-fibrotic genes and reduces collagen crosslinking in the ischemic heart.

**Anti-BMP1.3 antibody protects cardiomyocytes from ischemic damage**. By monitoring animals at multiple time points after MI, we observed that most parameters of cardiac function were preserved by the anti-BMP1.3 antibody early after LAD ligation, i.e. at 7 days, suggesting a direct protective effect of the treatment on cardiomyocyte survival, possibly independent from fibrosis prevention.

To verify this hypothesis, we assessed the extent of cell death early (2 days) after MI by staining both control and anti-BMP1.3 treated heart sections with TUNEL and the cardiomyocyte marker α-actinin. While control animals showed numerous TUNEL+ cardiomyocytes in the MI region, their number was significantly reduced by the anti-BMP1.3 treatment, already at low dose (Fig. 3a, b). Similar results were obtained by assessing the level of apoptosis, using anti-cleaved Caspase 3 immunostaining, as shown in Supplementary Fig. 5.

To confirm a direct, protective effect of the anti-BMP1.3 antibody on cardiac cells, we cultured primary cardiomyocytes and cardiac fibroblasts in a hypoxic chamber to mimic the shortage of oxygen following MI. Treatment of the cells with the anti-BMP1.3 antibody significantly reduced the number of apoptotic fibroblasts and cardiomyocytes (Fig. 3c, d). Conversely, the number of TUNEL+ cardiomyocytes, but not fibroblasts, was slightly increased by the addition of the recombinant BMP1.3 protein (Fig. 3e, f).

**BMP1.3 inhibition protects cardiomyocytes from hypoxia through the secretion of BMP5 by cardiac fibroblasts**. Members of the BMP family are known to control the survival of multiple cell types and tissues[36]. In particular BMP2, 4, 5, 6, 7 and 10 have been involved in cardiac development and/or in the compensatory response of the heart to ischemia[14,37]. We thus assessed whether inhibition of BMP1.3 affected the expression of these *Bmp* genes in both cardiomyocytes and cardiac fibroblasts, in either normoxia or hypoxia. Among all members, *Bmp2* and *Bmp5* expression was dramatically increased in hypoxic fibroblasts after treatment with the anti-BMP1.3 antibody. A similar, but lower effect was observed in hypoxic cardiomyocytes (Fig. 4a). This effect was still present, but less evident, in normoxic conditions (Supplementary Fig. 6a). Consistent with these data, significant upregulation of both Bmp2 and Bmp5 was also induced by specific siRNAs targeting BMP1.3 (Supplementary Fig. 7). Remarkable upregulation of both Bmp2 and Bmp5 was also observed in vivo, in the heart of mice exposed to MI and treated with the anti-BMP1.3 antibody, as shown by RT-PCR and immunohistochemistry in Fig. 4b, c, respectively.

To assess the involvement of *Bmp2* and *Bmp5* in the cardioprotective effect of the anti-BMP1.3 antibody, we exposed hypoxic cardiomyocytes to a medium enriched for the corresponding proteins, either individually or in combination. We found that media containing high levels of BMP2 and BMP5, either separately or combined, preserved cardiomyocyte viability in hypoxic conditions (Fig. 4d, e).

Next, we silenced the expression of *Bmp2* and *Bmp5* in primary cultures of fibroblasts and cardiomyocytes using specific siRNAs (Supplementary Fig. 6b). Interestingly, cardiomyocytes exposed to hypoxia and treated with the anti-BMP1.3 antibody were less protected, and their death was massively increased when *Bmp5* was silenced, either individually or in combination with *Bmp2*. (Fig. 4f, g). This suggested that Bmp5 was a key mediator in the protective activity exerted by the antibody.

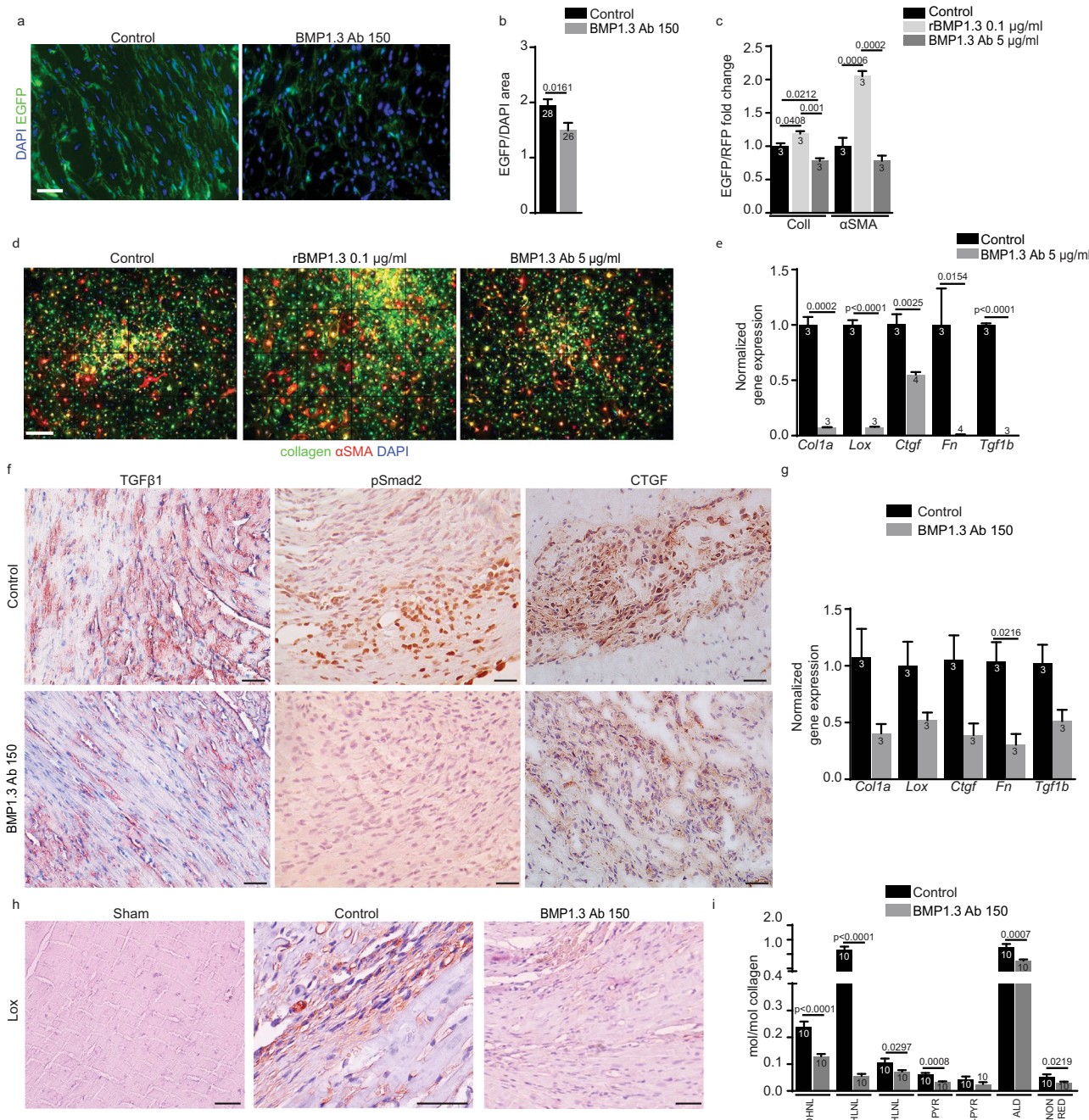

**Fig. 2 Modulation of TGFβ target genes and enzymes involved in fibrosis by anti-BMP1.3 antibody. a** Representative images of heart sections from COLL-EGFP mice after MI (control) and after treatment with anti-BMP1.3 antibody (150 μg/kg). Nuclei were stained with DAPI. **b** Quantification of the EGFP + area normalized on DAPI + area. **c** Quantification of both EGFP⁺ or RFP⁺ pixels in cultured cardiac fibroblasts from COLL-EGFP/SMA-RFP mice untreated (control), treated with human recombinant BMP1.3 (rhBMP1.3, 0.1 μg/ml) or anti-BMP1.3 antibody (BMP1.3 Ab, 5 μg/ml). **d** Representative images of cultured cardiac fibroblast from COLL-EGFP/SMA-RFP mice treated as indicated. **e** Quantification of the expression levels of *Col1a, Lox, Ctgf, Fn,* and *Tgf1b* genes normalized to *Gapdh* in cultured cardiac fibroblasts treated as described. **f** Representative images from tissue section of infarct area of animals untreated (control) or treated with anti-BMP1.3 antibody (150 μg/kg) stained with anti-Tgfβ1, anti-phospho-Smad2 and anti-CTGF antibodies (brown) and hematoxylin (violet) to visualize nuclei. **g** Quantification of the expression levels of *Col1a, Ctgf, Fn, Lox,* and *Tgf1b* genes normalized to *Gapdh* in hearts, either untreated (control) or treated with anti-BMP1.3 antibody (150 μg/kg). **h** Representative images from tissue section of the heart of sham animals or animals subjected to MI, either in the absence of treament (control) or treated with anti-BMP1.3 antibody (150 μg/kg), stained with anti-Lox antibodies (brown) and hematoxylin (violet) to visualize nuclei. **i** Biochemical analysis of collagen crosslinking in scar tissue dissected from hearts of untreated rats (control) or rats treated with anti-BMP1.3 antibody (150 μg/kg). DHNL dihydroxynorleucine, DHLNL dihydroxylysinonorleucine, HLNL hydroxylysinonorleucine, Pyr pyridinoline, d-Pyr deoxypyridinoline, ALD total aldehyde, NON RED non reduced aldehyde. Data in **b, c, e, g** and **i** are shown as mean ± s.e.m. Sample number is indicated inside or above each bar. Statistical significance was determined using unpaired, two-sided *t*-test in **b, e, g** and **i** or one-way ANOVA followed by Tukey's multiple comparison test in **c**. Scale bars in **a, f** and **h** is 100 μm, while in **d** is 500 μm. Source data are provided as a Source Data file.

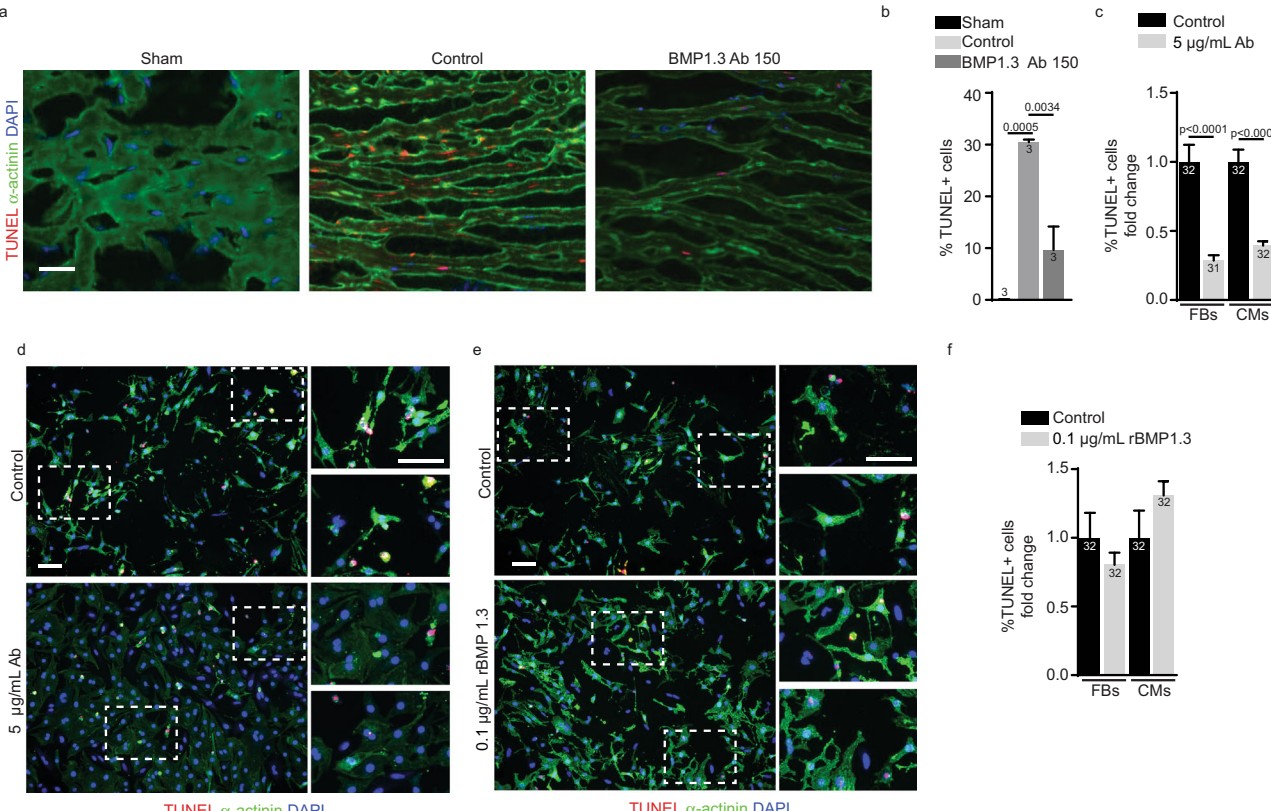

**Fig. 3 Modulation of cardiomyocyte apoptosis by anti-BMP1.3 antibody. a** Representative images of TUNEL and α-actinin staining on heart sections of mice untreated (sham) or injured by a MI in the absence of treatment (control) or after injection of anti-BMP1.3 antibody (150 μg/kg). Nuclei were stained with DAPI. **b** Quantification of dying cells in the infarcted area. **c** Quantification of TUNEL⁺ nuclei in primary cardiomyocytes and cardiac fibroblasts. **d** Representative images of primary cardiac cells, either untreated (control) or treated with anti-BMP1.3 antibody (5 μg/ml), stained with TUNEL and anti-α-actinin antibody. Nuclei were stained with DAPI. **e** Representative images of primary cardiac fibroblasts and cardiomyocytes, cultured in hypoxic conditions, either in the absence (control) or in the presence of recombinant BMP1.3 (0.1 μg/ml), stained with TUNEL and anti-α-actinin antibody. Nuclei were stained with DAPI. **f** Quantification of TUNEL⁺ nuclei in cardiomyocytes and cardiac fibroblasts. Data in **b**, **c**, and **f** are shown as mean ± s.e.m. Sample number is indicated inside or above each bar. Statistical significance was determined using one-way ANOVA followed by Tukey's multiple comparison test in **b**, **c** and **f**. Scale bars in **a**, **d** and **e** indicate 100 μm. Source data are provided as a Source Data file.

## Discussion

Here we show for the first time that a monoclonal antibody specifically inhibiting BMP1.3 reduces collagen deposition, protects cardiomyocytes from ischemic damage and preserves cardiac function following MI.

Our clinical data, showing increased BMP1.3 levels in the plasma of patients with MI, suggest that this isoform might play an important role in the response of the human heart to acute hypoxic damage. This is in line with previous evidence showing that BMP1 is upregulated after MI in rats[38] and that BMP1 inhibition by sFRP2 inhibits fibrosis and improves LV function after MI[38].

BMP1, and its soluble isoform BMP1.3, have been recently shown to be involved in tissue fibrosis in multiple organs. Their inhibition reduced fibrosis progression in both liver and kidney[27,29], indicating that the anti-BMP1.3 antibody targets common pathways that are relevant in the formation of fibrotic tissue in multiple organs. Fibroblast infiltration and collagen deposition are crucial during the first weeks after MI to preserve LV wall integrity and prevent cardiac rupture. However, excessive scar deposition leads to wall stiffness and progressive cardiac dysfunction, leading to heart failure. Therefore, we tested the capacity of anti-BMP1.3 therapy to interfere with excessive collagen deposition and scar extension in two rodent models of cardiac damage. In both models, BMP1.3 inhibition importantly

reduced scar size and resulted in improved cardiac function up to 2 months after injury. Importantly, the specific anti-BMP1.3 antibody was safely injected intravenously in vivo, with no apparent side effects. Thus, the potent anti-fibrotic effect was associated with a safety profile that was significantly higher compared to other strategies aimed at inhibiting scar formation after MI. For instance, we have previously shown that depletion of regulatory T cells reduces collagen deposition but leads to cardiac rupture[39] and TGFβ blockers are known to be fraught by major side effects, particularly in the heart. In particular, TGFβRI blockade using two different small receptor kinase inhibitors induced heart valve lesions in rats[40], and treatment with a pan-TGFβ neutralizing monoclonal antibody was associated with increased risk of bleeding and cardiac toxicity in both mice and monkeys[41].

In addition to its pro-collagenase activity, BMP1 is known to increase collagen production via TGFβ[42]. Our data indicate that the major mechanism of action of the anti-BMP1.3 monoclonal antibody is indeed inhibition of TGFβ pathway (Fig. 5). Multiple evidence supports this concept. Using primary fibroblasts from dual reporter mice, we showed that recombinant BMP1.3 promoted their activation into myofibroblasts, while anti-BMP1.3 antibody inhibited this process and downregulated the major TGFβ target genes involved in fibrosis. The capacity of the anti-BMP1.3 antibody to inhibit TGFβ pathway was also evident

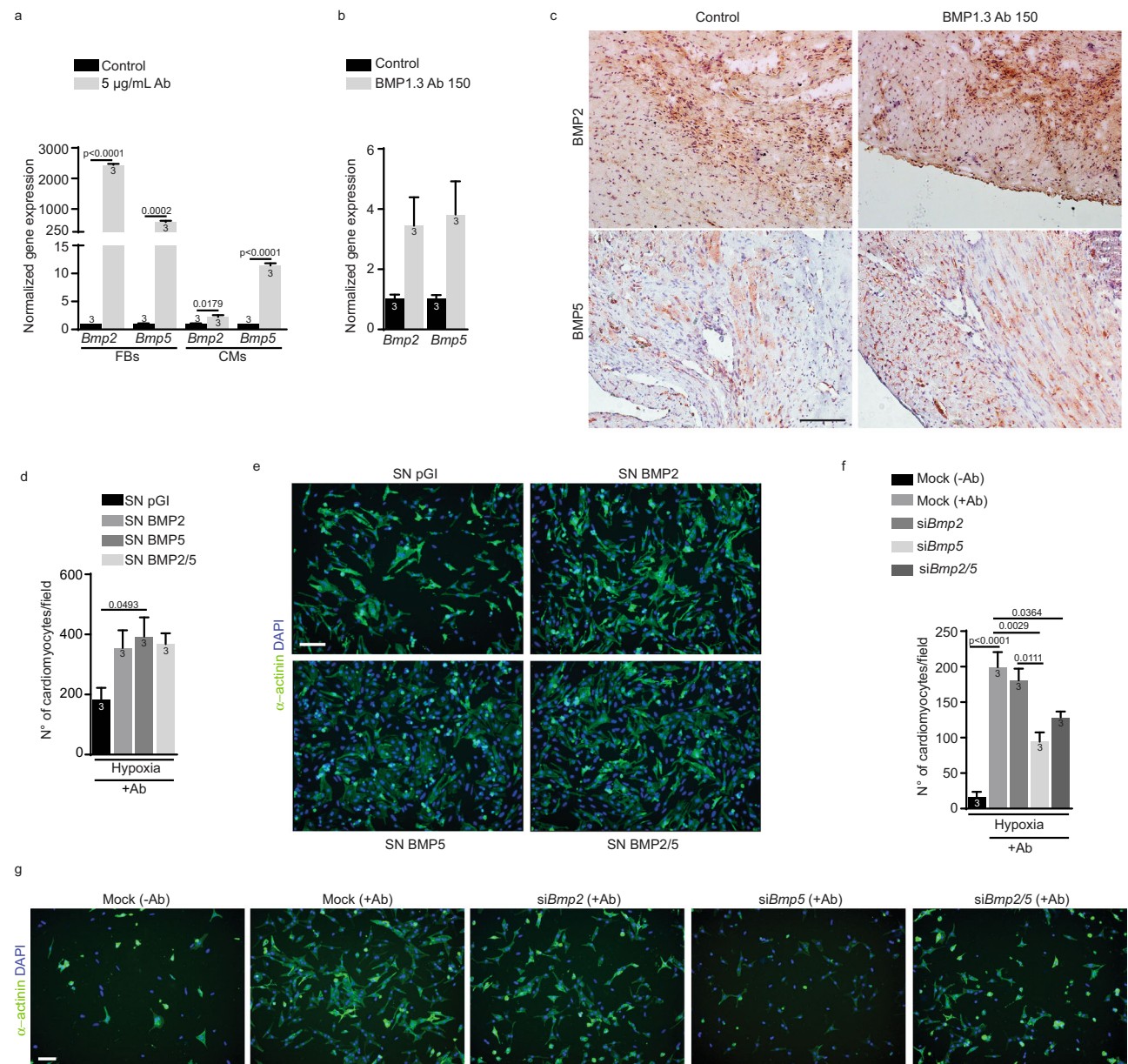

**Fig. 4 Bmp2 and Bmp5 are upregulated upon treatment with anti-BMP1.3 antibody and exert cardioprotective activity. a** Quantification of the expression levels of *Bmp2* and *Bmp5* in primary cardiac fibroblasts and cardiomyocytes cultured in hypoxic conditions in the absence of treatment (control) or treated with anti-BMP1.3 antibody (5 µg/ml), normalized to *Gapdh*. **b** Quantification of the expression levels of *Bmp2* and *Bmp5* in the heart of mice subjected to MI in the absence of treatment (control) or after injection of anti-BMP1.3 antibody (150 µg/kg). **c** Representative images from tissue section of infarct area of untreated rats (control) or treated with anti-BMP1.3 antibody (150 µg/kg) stained with either anti-BMP2 or anti-BMP5 antibody (brown) and hematoxylin (violet) to visualize nuclei. **d** Quantification of living cardiomyocytes cultured in hypoxic conditions expressed as number of cardiomyocytes per field and exposed to the indicated enriched supernatants collected from CHO transfected with control plasmid (pGI) or plasmid for the overexpression of BMP2 and/or BMP5. **e** Representative images of rat neonatal cardiomyocytes cultured in hypoxic conditions and exposed to the indicated treatments. Cardiomyocytes were stained with anti-α-actinin antibody and nuclei with DAPI. **f** Quantification of living cardiomyocytes cultured in hypoxic conditions and exposed to the indicated treatments. **g** Representative images of rat neonatal cardiomyocytes cultured in hypoxic conditions in the absence of treatment (−Ab) or treated with anti-BMP1.3 antibody (+Ab) and transfected with the indicated siRNAs (mock = scramble siBMP5). Data in **a**, **b**, **d** and **f** are shown as mean ± s.e.m. Sample number is indicated inside or above each bar. Statistical significance was determined using unpaired, two-sided *t*-test in **a**, **b** or one-way ANOVA followed by Dunnet's in **d** or Tukey's multiple comparison test in **f**. Scale bar in **c**, **e** and **g** is 100 µm. Source data are provided as a Source Data file.

in vivo, as shown by reduced levels of pSmad2 in the nuclei of fibroblasts within the scar tissue and, also in this case, significant downregulation of TGFβ target genes. Finally, recombinant BMP1.3 increased Smad2/3 promoter activity in a luciferase reporter assay, while anti-BMP1.3 antibody exerted opposite, inhibitory effect.

Among the TGFβ target genes is *Lox*. The role of Lox proteins in collagen crosslinking is well known[11] and was recently confirmed by the prevention of liver and lung fibrosis using anti-Loxl2 antibodies[11,43]. Our data indicate that anti-BMP1.3 inhibition significantly downregulated Lox expression both ex vivo and in vivo, and reduced collagen crosslinking in the heart after

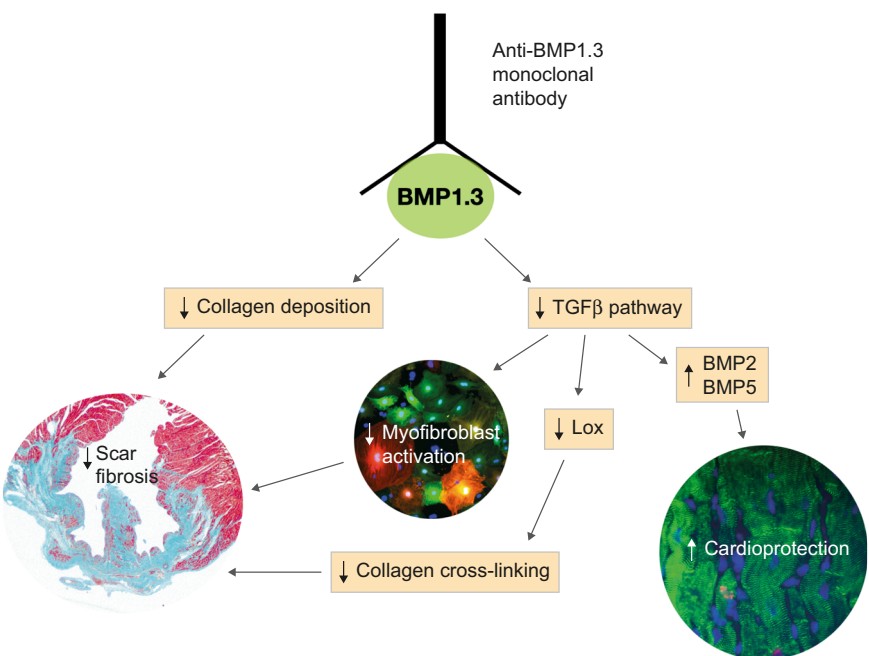

**Fig. 5 Mechanism of action of anti-BMP1.3 antibody.** The anti-BMP1.3 monoclonal antibody blocks the enzymatic activity of BMP1.3, exerting pleiotropic therapeutic effects after MI. On the one hand, BMP1.3 inhibition reduces collagen maturation. On the other hand, it results in TGFβ pathway inhibition, which in turns reduces myofibroblast activation, Lox expression and collagen cross-linking and increases the expression of cardioprotective BMPs.

MI. Reduced oxidation of lysyl and hydroxylysyl residues in both collagen and elastin chains render these molecules less prone to form covalent bonds with surrounding amino groups and other aldehydes[44–46]. This likely preserved the elasticity of the ECM and consequently improved cardiac function. Overall, these results show the ability of the anti-BMP1.3 antibody to reduce scar extension and collagen crosslinking, without the risk of compromising cardiac integrity and inducing cardiac rupture, as it might occur when cardiac fibrosis is excessively inhibited by alternative strategies[39].

Early apoptosis after MI importantly influences the subsequent remodeling of cardiac tissue. In line with this notion, treatment with the anti-BMP1.3 antibody significantly inhibited cardiomyocyte apoptosis in response to ischemic damage both in vivo and ex vivo. When primary cardiac cells were exposed to BMP1.3 they did not undergo massive cell death. This indicates that the antibody does not act by simply blocking a pro-apoptotic molecule, but its cardioprotective activity rather relies on more complex mechanisms. Having observed an effect of the anti-BMP1.3 antibody on both fibrosis and cardioprotection, we hypothesized the involvement of other BMPs, as they are known to control cardiac development and survival after ischemia[13,20,22,47,48]. Both cardiomyocytes, and more importantly, fibroblasts upregulated the expression of *Bmp2* and *Bmp5* in response to antibody treatment, likely because of the presence of Smad2/3 binding sites in their promoter sequences[49] (positions chr2: 133552604-133552616 and chr9: 75775051-75775064, respectively, according to UCSC Genome Browser). To understand whether BMPs are functionally involved in the cardioprotective effect of our antibody, we first tested the ability of these proteins to rescue cardiomyocyte viability in hypoxic conditions. BMP5 exerted the highest cardioprotective effect, either alone or in combination with BMP2. Next, we silenced their expression by specific siRNAs and again observed that the absence of BMP5 abolished the protective activity of the anti-BMP1.3 antibody. These findings collectively indicate that BMP5 mediates the early cardioprotective effect of the anti-BMP1.3 antibody, in line with its upregulation in vivo after MI and its pro-survival effect in neural crest

cells[50]. While BMP5 stands as the key mediator in cardioprotection, BMP2 might be responsible for reduced TGFβ expression. Indeed, in myogenic cells, BMP2 was shown to upregulate a series of transcription factors, including GLIS3[51], which usually acts as a negative regulator of transcription and has a specific binding site in the TGFβ1 promoter[52]. Although experimental evidence on the capacity of GLIS3 to inhibit TGFβ1 expression is lacking, this could represent one of the plausible mechanisms of action of the anti-BMP1.3 antibody, which could thus block TGFβ pathway at both transcriptional and post-translational levels.

Collectively, our data indicate that BMP1.3 inhibition exerts both anti-fibrotic and pro-survival effects on the heart after MI. To the best of our knowledge, this is the first example of a monoclonal antibody capable of modulating excessive scar formation in the infarcted heart. This concept could be extended to other diseases, in which fibrosis is a key pathogenetic mechanism, including chronic kidney, liver, lung diseases and muscular dystrophy. In all these conditions, reducing the deposition of excessive ECM via BMP1.3 inhibition might provide a therapeutic benefit.

## Methods

**Animals.** All animal experiments were conducted in accordance with guidelines from the Directive 2010/63/EU of the European Parliament on animal experimentation in compliance with European guidelines and International Laws and Policies (EC Council Directive 86/609, OJL 34, 12 December 1987) and were approved by the ICGEB Animal Welfare Board, the Ethical Committee and the Italian Ministry of Health (authorization number 1303/2015-PR) and by Institutional Animal Care Committee of Medical Faculty, University of Zagreb and Ministry of Agriculture, Republic of Croatia (authorization number 525-10/0255-15-6). Animals were housed in ventilated cages and subjected to a light dark cycle. Temperature was kept at 20–22 °C and relative humidity at 45–65%. Adult Sprague-Dawley rats, C57BL6 and αSMA-RFP/COLL-EGFP mice[53] were used for the in vivo procedures. Neonatal Sprague-Dawley rats and αSMA-RFP/COLL-EGFP[53] mice were used to obtain primary fibroblasts and cardiomyocytes for in vitro studies.

**BMP1.3 monoclonal antibody.** Purified BMP1.3 peptide (amino acids 972-986; RYTSTKFQDTLHSRK) was used as immunogen for the production of the

monoclonal antibody (mAb), using the standard hybridoma technique[25]. The specific monoclonal antibody cell line was produced by Promab Biotechnologies, Richmond, CA.

For the neutralization assay, recombinant dentin matrix protein 1 (DMP-1, 1 µg) was incubated with rBMP1.3 (125 ng) in reaction buffer (50 mM Tris-HCl, pH 7.5; 150 mM NaCl; 5 mM CaCl$_2$) with protease inhibitors (Sigma, P-2714) and incubated for 20 h at 37 °C. The reaction was stopped by adding 4X Lithium Dodecyl Sulfate (LDS) buffer containing β-mercaptoethanol and heating samples at 96 °C for 7 min. The BMP1.3 antibody (5 µg) was added to rBMP1.3 in reaction buffer and incubated for 3 h at 37 °C and then mixed with DMP-1, as before. All samples were resolved by SDS-PAGE and stained with Coomassie brilliant blue stain.

**BMP1.3 ELISA**. Human plasma samples were obtained from eight patients of both genders who had a recent MI and eight aged/gender matched healthy donors without comorbidities. Among AMI patients, 6 had hyperlipidemia and others did not have significant comorbidities. All participants were older than 40 years. Human samples were collected in compliance with national ethical regulations, upon approval by the Ethics Committee of the School of Medicine University of Zagreb (EC KBC 5367001) and informed consent by all patients. Participants did not receive any compensation.

Plasma levels of BMP1.3 were measured using the ELISA Human BMP1 Matched Antibody Pair Kit (Cloud-Clone PSA653Hu01) and additional reagents supplied in Antibody Pairs Support Pack (Cloud-Clone IS077). Both capture and detection antibodies were rabbit polyclonal antibodies raised to an immunogen sequence Glu610-Ser843, specific for the BMP1.3 isoform. Microtiter 96-well plates (Immulon-HB) were coated with 100 µl/well capture antibody (final concentration 4 µg/mL) and incubated overnight at 4 °C. Microtiter plates were then blocked with provided blocking buffer for 90 min at 37 °C. Recombinant hBMP1.3 (Cloud-Clone) was used as a standard for the calibration curve. Plasma samples were incubated for 1 h at 37 °C, followed by the addition of biotinylated detection antibody (final concentration 1 µg/mL) for 1 additional hour. HRP-streptavidin (Cloud-Clone) was diluted 1:100 and incubated 30 min at 37 °C. The reaction was visualized by the addition of the chromogenic substrate tetramethylbenzidine and stopped with 1 M H$_2$SO$_4$. Plates were washed with provided wash buffer after each step. All samples were analyzed in duplicate. Absorbance was measured with a microplate reader (Biotek, EL808B) at 450 nm.

Plasma levels of mouse BMP1.3 were measured using the Mouse BMP-1 ELISA kit (Novus Biologicals) which detects specifically BMP1.3 isoform. Plasma samples were obtained from mice 24 h after MI and from sham-operated animals.

**BMP1.3 antibody pharmacokinetics**. Single-dose pharmacokinetics of the monoclonal mouse anti-BMP1.3 antibody was performed after either intravenous (i.v.) or intraperitoneal (i.p.) injection in 2 months old female Sprague-Dawley rats weighing 300 g ($n = 3$ per administration route) at a dose of 50 µg/kg (injection volume 0.25 mL i.v., 0.5 mL i.p.). Blood samples were taken from tail vein immediately before and at 5, 15, 60, 180, and 360 min and then at 24, 48, 72, 144, 168, 192, 240, 312, and 384 h post-injection. Serum anti-BMP1.3 antibody concentrations were measured by indirect ELISA method with the injected mouse monoclonal antibody used as a standard for the calibration curves (concentration range 10 pg/mL to 10 µg/mL, diluted in 1% BSA in PBS and in rat serum pool). Microtiter plates were coated overnight at 4 °C with 150 ng/mL rhBMP1.3 diluted in carbonate buffer (15 mM Na$_2$CO$_3$, 35 mM NaHCO$_3$ buffer prepared without azide, pH 9.8) and then blocked with 5% sucrose and 1% BSA in PBS for 1 h at room temperature. Undiluted serum samples were incubated for 2 h at room temperature followed by the addition of biotinylated goat anti-rat IgG (R&D Systems), diluted 1:5000 as recommended by the manufacturer, for additional 2 h. Streptavidin-HRP (R&D Systems, 890803) was diluted 1:200 in 1% BSA in PBS and incubated for 20 min at room temperature. The reaction was visualized by the addition of chromogenic substrate (tetramethylbenzidine, TMB) and stopped with 1 M H$_2$SO$_4$. Plates were washed three times with PBS containing 0.05% Tween 20 (PBS-T) (pH 7.4) after each step. Absorbance was read at 450 nm using Absorbance Microplate Reader ELx 808TM. All serum samples were analyzed in duplicate.

**Rat and mouse models of cardiac fibrosis**. Cardiac injury was induced by systemic administration of isoproterenol in rats and by ligation of the LAD coronary artery in mice.

For rat experiments, 2 months old Sprague-Dawley animals, weighting ~280 g were used. Control and treated groups were injected subcutaneously with isoproterenol hydrochloride (Sigma Aldrich, Saint Louis, MO) at 85 mg/kg/day diluted in 2 mL of saline on two consecutive days with an interval of 24 h. Three experimental groups were established as follows: (1) sham—2 mL of saline on two consecutive days subcutaneously; (2) control—isoproterenol i.p.; (3) isoproterenol i.p. with anti-BMP1.3 antibody at 150 µg/kg, according to the protocol shown in Supplementary Fig. 2a. In a separate set of animals, the LAD was ligated in rats ($n = 20$) and the animals were treated with 150 µg/kg anti-BMP1.3 antibody according to the standard protocol. After 4 weeks rats were euthanized, and scars were dissected under microscope (Olympus BX53) and loupe (Olympus SZX10) and frozen at −80 °C for analysis of collagen crosslinking.

Mouse MI was induced by ligation of the LAD. Two months old, either CD-1 or αSMA-RFP/COLL-EGFP mice, weighing ~27 g, were used in these experiments. Mice were anesthetized with an i.p. injection of Xylazine 0.6 ml/kg and Ketamine (0.8 ml/kg). After left side thoracotomy, ligation of the LAD was performed with a 6–0 ethilon suture, 1–2 mm below the tip of the left auricle. The chest cavity and the skin were closed with 4–0 ethilon suture. Mice were intubated and ventilated during the procedure. Mice were randomized to four experimental groups: (1) sham—thoracotomy without LAD ligation; (2) control—MI without treatment; (3) MI treated with anti-BMP1.3 antibody (150 µg/kg); (4) MI treated with anti-BMP1.3 antibody (500 µg/kg). Therapy was administered as shown in Supplementary Fig. 2a. To evaluate the cardioprotective effect of anti-BMP1.3 antibody, mice were sacrificed at 2 days after LAD. To compare the efficacy of the anti-BMP1.3 antibody with alternative anti-fibrotic treatments, a new set of MI was performed in 16 mice, which were randomized to receive (1) anti-BMP1.3 antibody (150 µg/kg); (2) anti-TGFβ1 antibody (BioXCell, BE0057, 2 mg/kg)[54]; (3) SB-431542 (Sigma Aldrich, Saint Louis, MO, 10 mg/kg)[55], a chemical inhibitor specifically blocking the TGFβ receptors Alk4, Alk5, and Alk7, and (4) IgG1 antibody (150 µg/kg) as isotype control.

**Troponin T plasma levels**. Myocardial damage and necrosis were evaluated by measuring plasmatic levels of Troponin T on days 0, 1, 2, 3, 7 and 10, using a commercially available kit (Roche). Blood samples were drawn from the orbital plexus and collected in heparinized tubes. Samples were promptly centrifuged at $2000 \times g$ for 15 min prior to analysis.

**Primary culture of cardiomyocytes and cardiac fibroblasts**. Primary cardiomyocytes and fibroblasts were isolated as previously described[39]. Primary cell cultures were exposed to reduced oxygen levels (2%) for 48 h in a hypoxic chamber (Ruskinn IN VIVO2 200) and treated with either 5 µg/ml anti-BMP1.3 antibody or 0.1 µg/ml rBMP1.3. Alternatively, supernatants enriched with either BMP2 and/or BMP5 were produced as previously described[39], using plasmids encoding for mouse *Bmp2* and *Bmp5*. Enriched supernatants were added to the culture media (1:1 ratio with fresh medium) together, with anti-BMP1.3 antibody or rBMP1.3. Silencing of *Bmp1.3*, *Bmp2* and *Bmp5* in cardiomyocytes and fibroblasts was performed the day of plating using Lipofectamine RNAiMax (Thermofisher), following manufacturer instruction. A pool of siRNAs for *Bmp2* (siGenome Rat Bmp2 siRNA smartpool, Dharmacon, gene ref. 29373) and *Bmp5* (siGenome Rat Bmp5 siRNA smartpool, Dharmacon, gene ref. 315824) were used, while a specific siRNA for *Bmp1.3* (antisense strand 5' UGU GAU GCA GGU GAA AGC CUU 3') was custom designed and synthetized by Eurofin genomics.

Cells were fixed with 4% paraformaldehyde (PFA) in PBS at the end of the experiment and stained.

**Isolation of different cell types from ischemic mouse hearts**. Different cell types were purified from the heart 2 days after LAD ligation. Cardiomyocytes were isolated as described[56]. For endothelial cells, inflammatory cells and fibroblasts, hearts were digested to single cell suspension using the Skeletal Muscle dissociation kit (Miltenyi Biotech). Endothelial cells were positively selected using anti-CD31 coated magnetic beads (Miltenyi Biotech), following manufacturer instructions. CD31-depleted cells were subsequently incubated with anti-CD45 magnetic beads (Miltenyi Biotech) to separate inflammatory cells from cardiac fibroblasts. Purified cells were pelleted at 4 °C at $300 \times g$ for 5 min for RNA extraction.

**Echocardiography**. Echocardiography was performed on sham, control and anti-BMP1.3 antibody-treated rats and mice at days 7, 14, 30, and 60 using a VisualSonics Vevo 2100 ultrasound device with 55 MHz MicroScan transducer (in mice) and at day 30 using a 25 MHz MicroScan transducer (in rats). Mice were anesthetized with 1% isoflurane during the imaging. Parasternal short-axis and long-axis B-mode and M-mode views were recorded, as recommended by recent guidelines[57].

Left ventricular systolic function was assessed by ejection fraction, according to the formula: EF (%) = [(EDv − ESv)/EDv] × 100. Echocardiography was performed by two independent educated observers and mean value was taken.

**Gene expression analysis**. Total RNA from cultured cells and hearts was isolated using TRIzol (Invitrogen), following manufacturer instructions. Tissues were digested with tissue lysis buffer (20 mM Tris-HCl (pH 8.0), 1 mM CaCl$_2$, 0.1% sodium dodecyl sulfate, and 200 µg/ml proteinase K) and incubated at 55 °C for 12 h with shaking. cDNA was synthesized and amplified from 1 µg of total RNA using Super Script III First-Strand Synthesis System (Invitrogen). Gene expression of interest was measured using a LightCycler FastStart DNA Master SYBR Green kit in a LightCycler instrument (Roche Diagnostics), as described[58]. Values for specific genes were normalized to *Gapdh*, *18S* or *β actin* housekeeping controls. Results are represented as a fold change of the comparative expression level. The list of primers used for gene expression analysis is shown in Supplementary Table 3.

**Histology and immunostaining**. Heart samples were fixed in formalin solution and embedded in paraffin. For Sirius Red staining, 4 µm sections were stained with

hematoxylin-eosin and observed under light microscopy with camera. Sirius Red staining was quantified using S-form software.

Alternatively, hearts were cut on their sagittal plane and freshly frozen in liquid nitrogen-cooled isopentane. Frozen tissue was immersed in cryoprotective embedding medium (Killik, Bio Optica) and cut in 4 μm sections using a Leica cryostat.

Immunohistochemistry was performed after deparaffinization in xylene and rehydration in descending concentrations of ethanol with final incubation in PBS. Heat-induced epitope retrieval was performed in sodium citrate buffer pH 6.0 in a microwave oven for 15 min, followed by cooling for 30 min at room temperature. Either Mouse or Rabbit Specific HRP/AEC IHC Detection Kit—Micro-polymer (Abcam) was used in all procedures. To eliminate endogenous peroxidase activity and non-specific staining, sections were pretreated with Hydrogen Peroxide Block and Protein Block reagents from the same kit. Slides were incubated with mouse anti-BMP2 (1:200, Abcam ab6295), rabbit anti-BMP5 antibody (1:1000, Thermo Fisher Scientific PA5-78878), rabbit anti-TGFβ1 antibody (1:200, Abcam ab92486), mouse anti-Lox (1:100, R&D Systems MAB2639) and rabbit anti phosphorylated SMAD2 (1:100, kind gift of Peter ten Dijke) overnight at 4 °C in a moist chamber. Goat secondary antibody coupled with horseradish peroxidase (HRP) (Mouse and Rabbit Specific HRP/AEC IHC Detection Kit—Micro-polymer; Abcam: ab236467) was used and detected using 3-amino-9-ethylcarbazole (AEC) chromogen or diaminobenzidine (DAB, ImPACT DAB Substrate kit, Vector Laboratories). Slides were counterstained in hematoxylin and mounted using ImmunoHistoMount™ (Sigma-Aldrich, I1161); images were taken using the Olympus BX53 microscope. For immunofluorescence, cell death was evaluated using in Situ Cell Death Detection Kit, TMR red (Roche) on 5 μm thick section as previously described[39]. Slides were subsequently blocked in 0.1% Triton X-100 (Sigma), 10% horse serum (Gibco) solution in PBS and stained with anti-actinin antibody (1:3000, Abcam ab9465). After three washes in PBS, cells were incubated with an Alexa Fluor 488 anti-mouse secondary antibody diluted 1:500 in blocking solution for 1 h at room temperature. Nuclei were stained with DAPI. Alternatively, apoptosis was evaluated using antibodies against cleaved Caspase 3 (1:100, Cell Signaling 9661 s).

Primary cells were fixed with 4% PFA, permeabilized 10 min at room temperature with a 0.1% Triton X-100 (Sigma) solution in PBS, blocked with 2% BSA (Sigma) 0.2% Tween20 (Sigma) solution in PBS and stained with anti-α-actinin (1:200) in blocking solution. After three washes in PBS, cells were incubated with Alexa Fluor 488 anti-mouse secondary antibody (1:500, Invitrogen A-21202) in blocking solution for 1 h at room temperature. Nuclei were stained with DAPI.

**Collagen cross-link analysis.** Scar areas of control ($n = 6$) and treated rats ($n = 6$) were quantitatively analyzed following mincing and demineralization with EDTA (6 h), reduced with $NaB_3H_4$, hydrolyzed with 6 N HCl and subjected to amino acid and cross-link analysis[59]. Reducible cross-links (dehydro-dihydroxy-ylysinonorleucine (deH-DHLNL)/its ketoamine) and a major aldehyde (hydro-xylysine-aldehyde) were analyzed as their reduced forms; i.e. DHLNL, HLNL and DHNL, respectively. The levels of non-reducible cross-links, pyridinoline (Pyr) and deoxypyridinoline (d-Pyr) and reducible cross-links/aldehyde were expressed as moles of cross-link/mole of collagen. Total aldehyde involved in cross-linking was calculated by (DHNL + DHLNL + HLNL + 2xPyr + 2xd-Pyr). Maturation of cross-links was assessed by Pyr + d-Pyr/DHLNL + HLNL.

**Western blot.** To assess the level of Smad2/3 phosphorylation, fibroblasts were lysed using RIPA buffer (1% NP40, 10 mM Tris-HCl pH 7.4, 5 mM EDTA, 150 mM NaCl, 0.1% SDS, 0.1% sodium deoxycholate, 2 mM sodium orthovanadate, 10 mM sodium fluoride) supplemented with protease and phosphatase inhibitors (Sigma). Samples were denatured by boiling for 5 min at 95° degrees in Laemmli buffer and separated by SDS-Page using a 10% bis-tris acrylamide gel. Proteins were blotted on a nitrocellulose membrane using the semi-dry Power Blotter system (Invitrogen). Membranes were blocked in 5% BSA (for pSMAD2/3) or 5% skimmed milk (for total SMAD2/3) in TBS 0.1% Tween20 (TBST) for 30 min at room temperature, then incubated overnight at 4 °C with rabbit anti-Smad2/3 (Cell Signaling 8685) or rabbit anti-phospho-Smad2 (Ser465/467)/Smad3 (Ser423/425) (Cell Signaling 8828), diluted 1:1000 in their blocking solution. After five washes with TBST, membranes were incubated with goat anti-rabbit HRP-conjugated antibody (DAKO, p0448), diluted 1:2000 in their blocking solution for 1 h at room temperature. After five washes with TBST, membranes were incubated with SuperSignal West Pico PLUS Chemiluminescent Substrate (Thermoscientific) and chemiluminescence was detected using Chemidoc Touch imaging system (Promega).

**Luciferase assay.** HEK293T cells stably expressing the TGFβ-sensitive (CAGA)12-luciferase reporter (pGL3 CAGA12 LC + PuroR-P2A-Renilla-T2A-EGFP)[60] were cultured in serum-free DMEM. After 6 h starvation, 2.5 pg/ml of human recombinant TGFβ1 (Peprotech) was added to the medium, either alone or in combination with 5 μg/ml anti-BMP1.3 antibody or 0.1 μg/ml rBMP1.3. After additional 17 h, cells were lysed with Glo-lysis buffer (Promega) and assayed using the Dual-Glo luciferase kit (Promega) on the Wallac Envision plate reader (Perkin Elmer).

**Statistical analysis.** Data were collected using Microsoft Excel 2009 and analyzed with SPSS software (iBM SPSS Statistics version 2.2.0., New York, USA) or Graphpad Prism 6.0 (Graphpad software). Values are reported as mean ± S.E.M. Gaussian distribution was assessed and the appropriate Student's t-test of

Mann–Whitney U test was used to compare two groups, with one-way ANOVA followed by the appropriate post hoc test to compare more than two groups. Two-way ANOVA for repeated measures was used for comparisons between groups over time. Sample size was described in each figure legend. P values ≤ 0.05 were considered statistically significant.

**Reporting summary**. Further information on research design is available in the Nature Research Reporting Summary linked to this article.

## Data availability
The data supporting the findings from this study are available within the manuscript and the Supplementary information. Source data are provided with this paper.

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

## Acknowledgements

This work was supported by project INCardio funded by the European Regional Development Fund and Interreg V-A Italy-Austria 2014-2020 to S.Z. and M.M. and by the Scientific Center of Excellence for Reproductive and Regenerative Medicine (project "Reproductive and regenerative medicine—exploration of new platforms and potentials" GA KK01.1.1.01.0008 funded by the EU through the ERDF) to S.V.

## Author contributions

S.V. and I.D.C. had the idea and prompted the development of the anti-BMP 1.3 monoclonal antibodies; S.Vo. analyzed the in vivo effect of the antibody; V.K. and R.C. performed gene expression analysis; V.M. and S.M. took care of cell cultures; V.R., M.Mi., B.B., D.D.B. and T.B.N. characterized the antibody in vitro; M.G. and M.Ma. provided critical insights; S.Z. and A.C. coordinated the study and wrote the manuscript.

## Competing interests

S.V. is inventor in the following patents related to anti-fibrotic effect of BMP1.3 issued to School of Medicine University of Zagreb: US 7850964, JP 5016041, AU 2007275580, EP2518496, CA 2658582; JP 6133402, New Zealand 631639. The other authors do not have any competing interest.
