## [Peer Review File · Nature Communications]

REVIEWER COMMENTS

Reviewer #1 (Remarks to the Author):

Dumic-Cule, et al. describe the generation and application of an anti-BMP1 antibody to reduce cardiac fibrosis. They provide evidence that BMP1 inhibition enhances cardiomyocyte survival and reduces fibrosis. Overall, the manuscript is missing significant details about the antibody and the novelty compared other known fibrotic inhibitors (TGFb, LOX) or the authors previous work (ref 27 and 29) is not established.

1) The manuscript needs significantly more details and characterization of the antibody. The authors should provide data that confirms the specificity to BMP1 and neutralization properties of the antibody. What isotope is used for in vivo studies?

2) Why was no isotype control antibody used for in vivo studies?

3) BMP1.3 ELISA method description appears incomplete. Recombinant hBMP1 is capture via the R&D anti-BMP1 antibody, but how is it detect? What antibody and concentration were used?

4) To extend the novelty of the approach, the authors should directly compare their anti-BMP1 to other inhibitors of fibrosis such as TGFb or LOX. It's not currently clear what the advantage of BMP1 inhibition is to targeting other pro-fibrotic factors.

Reviewer #2 (Remarks to the Author):

NCOMMS-21-00704

Bone morphogenetic protein 1.3 inhibition supports cardiomyocyte survival and decrease scar formation after myocardial infarction

In this manuscript, Domic-Cule et al demonstrated that BMP1.3 levels were significantly increased in patients with MI compared to healthy individuals. Utilizing rodent models of MI, the authors showed that administration of a mouse monoclonal antibody against BMP1.3 resulted in reduced cardiac fibrosis and preserved cardiac functions. Anti-BMP1.3 antibody treatment decreased the expressions of pro-fibrotic genes (Col1a, Tgf1b and Lox) and inhibited collagen crosslinking. The authors further claimed that this treatment protects cardiomyocytes and cardiac fibroblasts from ischemic damage through the enhanced secretion of BMP5.

This study represents a large body of work, and the in vivo results are clinically interesting. However, molecular mechanisms underlying the cardioprotective effects evoked by BMP1.3 inhibition are not convincing.

1) Although the previous study demonstrated that BMP1 was upregulated after MI in rats (PNAS 107, 21110-21115, 2010), the authors should examine whether the plasma levels of BMP1.3 are also increased in their rodent model.

2) The in-house developed ELISA was used in Fig. 1a. The results of validation experiments for this ELISA should be provided.

3) If the increased plasma levels of BMP1.3 are observed in the rodent models used in this manuscript, which cell-type or tissue is the main source of the circulating BMP1.3?

4) The authors showed that anti-BMP1.3 antibody treatment decreased the expression of Tgf1b. However, the previous study suggested that BMP1 regulates the activity of TGFb by a posttranslational mechanism (J Cell Biol 175:111-120, 2006). How does the BMP1.3 inhibition decrease the expression of Tgf1b? Plausible explanation for this discrepancy should be provided.

5) In Fig. 4, the authors showed that anti-BMP1.3 antibody treatment increased the expression of Bmp2 and Bmp5. A molecular mechanism that links BMP1.3 inhibition and the increased expressions of Bmp2 and Bmp5 should be provided.

Reviewer #3 (Remarks to the Author):

Review NCOMMS-21-00704

General comments:

This is a very interesting study by a well-established group in the field of BMPs and fibrosis research, providing evidence that an anti-BMP1.3 antibody, at least in part through modulation of BMP/TGF- β balance, might have significant potential for development into clinical application for an important unmet need. A nice array of analyses is presented, ranging from functional in vivo data to gene expression, and posttranslational modification relevant to fibrosis.

The experimental data generally look sound, but seem at times incomplete, and some further detail would be appreciated.

Also, it would be appreciated if it could be emphasized where observations in this study might address final common pathways previously interrogated in other organ systems and where benefits of the BMP1.3-antibody exceed or differ from those of other interventions in previously published experimental MI. studies.

1. In this manuscript I have not found experiments showing rBMP1.3 effect is neutralized by Ab, neither that an alternative way of silencing BMP1.3 is congruent with Ab effect. For this, however, reference to previous experiments addressing fibrosis in other organ systems might suffice.
2. Is hypoxia itself, or the reperfusion hyperoxia driver of apoptosis? Moreover, in a concept centering around cell death driving fibrosis, other forms of regulated cell death (e.g. necroptosis) might be more relevant than “clean” apoptosis. In this sense, the study fails to clearly link improved survival of injured myocardial cells and mitigation of fibrosis, the latter even being presented in the introduction as a more or less autonomously progressive process.
3. Several of the datasets appear to be incomplete, e.g. lacking either α -Actinin or TUNEL, and inclusion of siBMP1.3 and/or rBMP1.3, and shams for normalization (see specific comments below)
4. Since BMPs vary in their in vivo expression profiles, potency and receptor preferences, the statement “Of all BMPs tested, BMP2 was remarkably upregulated ..” is a bit cryptic. Please specify which BMPs were tested and what is meant by “remarkably” as compared to other BMPs? Was BMP2 and -5 upregulation also found in Ab-treated (MI) hearts?
5. More extensive in vivo analyses would be appreciated, e.g. analysis of Ab treatment effect on tissue mRNAs for e.g. BMP1.3 and/or BMP2 and -5.
6. It would be helpful if analysis could be shown of the in vivo impact of BMP1.3 and the -Ab on BMP/TGF- β signaling balance, e.g. IHC for pSmad1,5,8/pSmad2,3, or ISH/qPCR for target-gene expression, preferably (also) in vivo.

7. The addition of one or two representative human sample(s) including healthy myocardium, AMI border zone, and –scar, could tremendously increase the translational relevance of the experimental data. These could be analyzed for mRNA and protein after microdissection, and/or stained for stained (by IHC or ISH) for BMP1.3 expression, α SMA, BMP2 and -5 expression and for pSmads 2,3, and -1,5,8.

Details:

Figure 1:

A-C N=8, while D-E show only 3, F N=4,. Selection?

E: 150 vs 500 doesn't look like reflecting the hearts that are shown

G-H: fractional shortening and ejection fraction increased by AMI and not altered by Ab? So where is the functional benefit?

Figure 2:

C: rBMP1.3 has much more impact on α SMA induction than on Collagen, but inhibitory effect of the Ab seems similar for both. Please comment.

E: Why a different layout for E than for C?

Remarkable that there is no effect of rBMP1.3 (high "baseline?"), but a profound effect of Ab.

Also remarkable that the Ab causes a total shut down of TGF1b mRNA expression: what could be the pathway, considering the multiplicity of factors that can induce TGF1b transcription?

G: Showing shams would be helpful for appreciation of the magnitude of (rescue) effect

Are DPYR and ALD known to be less dependent on BMP1.3/ Lox than other crosslinks?

Fig 3

ABC: How to understand that TUNEL is 30 –fold lower than α Actinin? Just a narrower window for detection or another stage of the process, and is this relevant to MOA of (anti-)BMP1.3?

What is the intervention point of BMP1.3 in apoptosis induction? Is this independent of the apoptosis induction pathway?

DEF: Why no quantification of α Actinin (cf Fig 4 D)?

F: Is this spontaneous apoptosis in culture?

Fig 4:

b: Why focus on BMP 5? BMP2 is 10x higher (in FB) in CM BMP5 5x higher. Page 13-14 (Discussion) cites some in vitro observations, but what is conceptually more important/interesting? How is expression of different BMPs in vivo?

D: Here no TUNEL? cf Fig 3D-F

E: siBMP1.3 would have been an interesting too!

F: Living, vs apoptotic in previous

Sincerely,

Roel Goldschmeding

Manuscript NCOMMS-21-00704

Bone morphogenetic protein 1.3 inhibition decreases scar formation and supports cardiomyocyte survival after myocardial infarction

Response to Reviewers.

We are grateful to Reviewers for their constructive comments and suggestions, which we believe have greatly improved the quality of our manuscripts. Reviewer's comments are in black, our response in blue.

Reviewer #1 (Remarks to the Author):

Dumic-Cule, et al. describe the generation and application of an anti-BMP1 antibody to reduce cardiac fibrosis. They provide evidence that BMP1 inhibition enhances cardiomyocyte survival and reduces fibrosis. Overall, the manuscript is missing significant details about the antibody and the novelty compared other known fibrotic inhibitors (TGF β , LOX) or the authors previous work (ref 27 and 29) is not established.

1) The manuscript needs significantly more details and characterization of the antibody. The authors should provide data that confirms the specificity to BMP1 and neutralization properties of the antibody. What isotope is used for in vivo studies?

We agree with the Reviewer on the relevance of showing the capacity of the antibody to specifically neutralize BMP1.3 activity. To address this issue, we now show the results of a neutralization assay, in which BMP1.3 cleaves its substrate DMP-1 and this activity is completely inhibited by anti-BMP1.3 antibody. These results are included in Supplementary Figure 1a.

For in vivo studies, we have used IgG1, consistent with the new data, which include an isotype control group (Supplementary Figure 3).

2) Why was no isotype control antibody used for in vivo studies?

We thank the Reviewer for this question. In our original study we were more interested in assessing the therapeutic efficacy than specificity of the anti-BMP1.3 antibody and therefore we omitted this control. Following Reviewer's suggestion, in the new set of experiments, we have now incorporated an isotype control group for in vivo studies (Supplementary Figure 3).

3) BMP1.3 ELISA method description appears incomplete. Recombinant hBMP1 is capture via the R&D anti-BMP1 antibody, but how is it detect? What antibody and concentration were used?

Following Reviewer's recommendation, we have now inserted more details on our ELISA in both main and supplementary methods. Briefly, for both capture and detection, we have used a rabbit polyclonal antibody, raised to the epitope Glu610-Ser843, specific for BMP1.3 isoform, used at 4 μ g/mL for capture and at 1 μ g/mL for detection.

4) To extend the novelty of the approach, the authors should directly compare their anti-BMP1 to other inhibitors of fibrosis such as TGF β or LOX. It's not currently clear what the advantage of BMP1 inhibition is to targeting other pro-fibrotic factors.

Following Reviewer's suggestion, we performed a new series of MI, in which we have compared the effect of the anti-BMP1.3 antibody with two alternative approaches to inhibit TGF β signaling, namely an anti-TGF β 1 antibody and a chemical inhibitor of the major TGF β receptors Alk4, Alk5 and Alk7. These new data, showing the superior efficacy of the anti-BMP1.3 antibody, are shown in Supplementary Figure 3.

Reviewer #2 (Remarks to the Author):

NCOMMS-21-00704

Bone morphogenetic protein 1.3 inhibition supports cardiomyocyte survival and decrease scar formation after myocardial infarction

In this manuscript, Dumic-Cule et al demonstrated that BMP1.3 levels were significantly increased in patients with MI compared to healthy individuals. Utilizing rodent models of MI, the authors showed that administration of a mouse monoclonal antibody against BMP1.3 resulted in reduced cardiac fibrosis and preserved cardiac functions. Anti-BMP1.3 antibody treatment decreased the expressions of pro-fibrotic genes (Col1a, Tgf1b and Lox) and inhibited collagen crosslinking. The authors further claimed that this treatment protects cardiomyocytes and cardiac fibroblasts from ischemic damage through the enhanced secretion of BMP5.

This study represents a large body of work, and the in vivo results are clinically interesting. However, molecular mechanisms underlying the cardioprotective effects evoked by BMP1.3 inhibition are not convincing.

1) Although the previous study demonstrated that BMP1 was upregulated after MI in rats (PNAS 107, 21110-21115, 2010), the authors should examine whether the plasma levels of BMP1.3 are also increased in their rodent model.

Following Reviewer's suggestion, we have assessed circulating levels of BMP1.3 in mice and confirmed they are increased after MI. These data are now included in Figure 1c.

2) The in-house developed ELISA was used in Fig. 1a. The results of validation experiments for this ELISA should be provided.

Following Reviewer's recommendation, we have now inserted more details on ELISA methodology and validation in both main and supplementary methods.

3) If the increased plasma levels of BMP1.3 are observed in the rodent models used in this manuscript, which cell-type or tissue is the main source of the circulating BMP1.3?

We thank the Reviewer for this question, which we have addressed by measuring BMP1.3 expression levels in total hearts after MI (Figure 1d). We also purified the main cardiac cell types (cardiomyocytes, endothelial cells, fibroblasts and inflammatory cells) from murine hearts after MI and assessed BMP1.3 levels by RT-PCR. We found that fibroblasts are the main source of BMP1.3. while both cardiomyocytes and endothelial cells express it at lower levels (Figure 1e).

4) The authors showed that anti-BMP1.3 antibody treatment decreased the expression of Tgfb. However, the previous study suggested that BMP1 regulates the activity of TGFβ by a posttranslational mechanism (J Cell Biol 175:111-120, 2006). How does the BMP1.3 inhibition decrease the expression of Tgfb? Plausible explanation for this discrepancy should be provided.

We agree with the Reviewer that the most plausible mechanism by which BMP1.3 can control TGFβ activity is by post-translational modification. Indeed, while ample literature exists on the post-translational control of TGFβ activity, scant information is available on its transcriptional regulation. Our data indicate that our anti-BMP1.3 antibody induces the expression of BMP2 and BMP5 in primary cultures of cardiomyocytes and fibroblasts. In myogenic cells, BMP2 has been shown to up-regulate a series of transcription factors, including GLIS3,¹ which usually acts as a negative regulator of transcription and has a specific binding site in the TGFβ promoter². This could represent a plausible mechanism by which anti-BMP1.3 antibody could inhibit TGFβ expression, meaning that it may block TGFβ pathway at both transcriptional and post-translational levels. This concept is now included in the Discussion.

5) In Fig. 4, the authors showed that anti-BMP1.3 antibody treatment increased the expression of Bmp2 and Bmp5. A molecular mechanism that links BMP1.3 inhibition and the increased expressions of Bmp2 and Bmp5 should be provided.

Consistent with our response to the previous point, the most plausible mechanism by which BMP1.3 can alter gene expression of a variety of target genes is through post-translational modification of TGFβ and modulation of downstream signaling pathway. Indeed, high levels of TGFβ are known to repress Bmp2 expression³ and multiple Smad2/3 binding sites are present in both Bmp2³ and Bmp5 promoters (positions: chr2:133552604 – 133552616 and chr9:75775051-75775064, respectively, according to UCSC Genome Browser). The new version of the manuscript provides convincing evidence that TGFβ pathway is inhibited by anti-BMP1.3 antibody both in cells and in vivo and this mechanism is now included in the Discussion.

Reviewer #3 (Remarks to the Author):

Review NCOMMS-21-00704

General comments:

This is a very interesting study by a well-established group in the field of BMPs and fibrosis research, providing evidence that an anti-BMP1.3 antibody, at least in part through modulation of BMP/TGF-β balance, might have significant potential for development into clinical application for an important unmet need. A nice array of analyses is presented, ranging from functional in vivo data to gene expression, and posttranslational modification relevant to fibrosis.

The experimental data generally look sound, but seem at times incomplete, and some further detail would be appreciated.

Also, it would be appreciated if it could be emphasized where observations in this study might address final common pathways previously interrogated in other organ systems and where benefits of the BMP1.3-antibody exceed or differ from those of other interventions in previously published experimental MI studies.

We thank the Reviewer for this suggestion. We do believe that our anti-BMP1.3 antibody acts on common pathways (i.e. TGFβ-induced activation of myofibroblasts, collagen cross-linking) that are relevant in the development of fibrosis in other organs, consistent with its efficacy in animal models of renal⁴ and liver fibrosis⁵. This concept is now emphasized in the Discussion. When compared to other strategies aimed at reducing fibrotic scar after MI, our antibody seems to have minimal side effects. For instance, in our experience T-reg depletion did reduce collagen deposition but often led to cardiac rupture⁶, and TGFβ-blockers are fraught by major side

effects, particularly in the heart. TGF β RI blockade using two different small receptor kinase inhibitors induced heart valve lesions in rats⁷ and treatment with a pan-TGF β neutralizing monoclonal antibody was associated with an increased risk of bleeding and cardiac toxicity in mice and monkeys⁸. In addition to the higher safety profile, which is emphasized in the Discussion, the superior efficacy of our anti-BMP1.3 antibody compared to other TGF β in preserving cardiac function is also supported by the new set of data, included in Supplementary Figure 3.

1. In this manuscript I have not found experiments showing rBMP1.3 effect is neutralized by Ab, neither that an alternative way of silencing BMP1.3 is congruent with Ab effect. For this, however, reference to previous experiments addressing fibrosis in other organ systems might suffice.

As the same criticism was also raised by Reviewer 1 and we agree on the relevance of showing the neutralizing properties of our antibody, we now show the results of a neutralization assay, in which BMP1.3 cleaves its substrate DMP-1, and this activity is completely inhibited by anti-BMP1.3 antibody. These results are included in Supplementary Figure 1a.

We also silenced BMP1.3 using specific siRNAs, which led to 50% reduction in expression. We did not have many options for siRNA design, as to be specific the siRNA had to match the sequence specific for BMP1.3 long isoform. While this partial BMP1-3 silencing recapitulated the effect of the anti-BMP1.3 antibody on Bmp2 and Bmp5 up-regulation (albeit to a lower extent compared to the Ab, shown in Supplementary Figure 7), it induced only a trend toward increased cardiomyocyte survival, as shown by the images below. Since we could not obtain significant differences in cardiomyocyte survival, we would prefer not to include these data in the manuscript.

2. Is hypoxia itself, or the reperfusion hyperoxia driver of apoptosis? Moreover, in a concept centering around cell death driving fibrosis, other forms of regulated cell death (e.g. necroptosis) might be more relevant than “clean” apoptosis. In this sense, the study fails to clearly link improved survival of injured myocardial cells and mitigation of fibrosis, the latter even being presented in the introduction as a more or less autonomously progressive process.

While we would tend to exclude that reperfusion could be the trigger of cell death in our model, as we performed permanent ligation of the LAD, without re-opening of the vessel, we have now stained the same heart samples with anti-cleaved Caspase 3 antibodies, which more specifically label apoptotic cells. These results are reported in Supplementary Figure 5 and are consistent with apoptotic death of cardiomyocytes upon prolonged hypoxia, in the absence of reperfusion-driven inflammation.

3. Several of the datasets appear to be incomplete, e.g. lacking either α -Actinin or TUNEL, and inclusion of siBMP1.3 and/or rBMP1.3, and shams for normalization (see specific comments below)

4. Since BMPs vary in their *in vivo* expression profiles, potency and receptor preferences, the statement “Of all BMPs tested, BMP2 was remarkably upregulated ..” is a bit cryptic. Please specify which BMPs were tested and what is meant by “remarkably” as compared to other BMPs? Was BMP2 and -5 upregulation also found in Ab-treated (MI) hearts?

5. More extensive *in vivo* analyses would be appreciated, e.g. analysis of Ab treatment effect on tissue mRNAs for e.g. BMP1.3 and/or BMP2 and -5.

Following Reviewer’s recommendation, we have inserted a paragraph to specify that we have analyzed the expression of the BMPs known to be involved in cardiac development and ischemic disease, and that only Bmp2 and 5 were found to be upregulated. We also analyzed the expression levels *in vivo* by both real-time PCR and immunohistochemistry and found a significant up-regulation of both Bmp2 and 5 in hearts of mice treated with our antibody. These new data are now included in Figure 4.

6. It would be helpful if analysis could be shown of the *in vivo* impact of BMP1.3 and the –Ab on BMP/TGF- β signaling balance, e.g. IHC for pSmad1,5,8/pSmad2,3, or ISH/qPCR for target-gene expression, preferably (also) *in vivo*.

We thank the Reviewer for this suggestion. A detailed analysis of TGF β signaling upon anti-BMP1.3 treatment is now shown both *ex vivo* and *in vivo*, in Figure 2 and Supplementary Figure 4, and includes WB and IHC for pSmad2, luciferase assay for SMAD2/3 promoter activity, and analysis of TGF β target gene expression

(collagen1, CTGF, fibronectin and Ix) by RT-PCR and IHC. We also stained the same samples for pSmad1,5,8 and showed some positive cells particularly evident upon anti-BMP1.3 treatment. However, as this staining also resulted in some positive signal in the cytoplasm of cardiomyocytes, we did not feel confident to include these data in the manuscript (some representative images are shown here).

7. The addition of one or two representative human sample(s) including healthy myocardium, AMI border zone, and –scar, could tremendously increase the translational relevance of the experimental data. These could be analyzed for mRNA and protein after microdissection, and/or stained for stained (by IHC or ISH) for BMP1.3 expression, α SMA, BMP2 and -5 expression and for pSmads 2,3, and -1,5,8.

Following the Reviewer's suggestion, we have checked the expression of BMP1.3 in human hearts affected by ischemic disease and healthy controls. Levels of BMP1.3 appear to be indeed increased after AMI, consistent with increased circulating levels of the protein. These real-time PCR data are now included in Figure 1b (we could not provide IHC because of lack of suitable antibodies).

We do not fully understand the rationale for assessing levels of SMA, BMP2, BMP5 and pSmad in these samples, as human patients have not been treated with the Ab. On the other hand, there is ample literature investigating TGF β pathway after MI in both animal models and in humans^{9,10}.

Details:

Figure 1:

A-C N=8, while D-E show only 3, F N=4,.

As mentioned in the text, rat experiments included 8 animals per group, while mouse experiments shown in Figure 1 included 4 animals per group. A few representative images for each group are shown for space constraint.

E: 150 vs 500 doesn't look like reflecting the hearts that are shown

We apologize for the lack of clarity. Quantification reflects the real amount of fibrosis in each heart. We have now included a selection of images that better correspond to the quantitative analysis.

G-H: fractional shortening and ejection fraction increased by AMI and not altered by Ab? So where is the functional benefit?

We apologize for the confusion, due to our mistake in the legend. This has now been corrected.

Figure 2:

C: rBMP1.3 has much more impact on α SMA induction than on Collagen, but inhibitory effect of the Ab seems similar for both. Please comment.

We thank the Reviewer for this interesting observation. Cultured fibroblasts spontaneously undergo differentiation into myofibroblasts when kept in culture for a few days and this is reasonably due to a variety of stimuli, independent from BMP1.3. Thus, we expect the Ab to have a minimal activity on basal activation, while addition of the recombinant protein at a high dose potently induces myofibroblast activation.

E: Why a different layout for E than for C?

We apologize for the confusion. The layout is now uniform.

Remarkable that there is no effect of rBMP1.3 (high "baseline?"), but a profound effect of Ab.

As mentioned before, cultured primary fibroblasts gets spontaneously activated, thus we can hypothesize a high level of BMP1.3 expression. The partial discrepancy with the quantification of reporter gene expression, might be due to the fact that in reporter mice only the main promoter sequence is cloned and it presumably does not fully reproduce the whole genomic context of the real endogenous promoter. To avoid confusion, we have now removed data on the effect of the recombinant protein from this panel, also for consistency with the following panels, which show a parallel decrease of the TGF β pathway upon anti-BMP1.3 Ab treatment both in vitro and in vivo.

Also remarkable that the Ab causes a total shut down of TGF1b mRNA expression: what could be the pathway, considering the multiplicity of factors that can induce TGF1b transcription?

As also pointed out by Reviewer 2, the most plausible mechanisms by which BMP1.3 can control TGF β activity is by post-translational modification. Indeed, while ample literature exists on the post-translational control of TGF β activity, scant information is available on its transcriptional regulation. Our data indicate that our anti-BMP1.3 antibody induces the expression of BMP2 and BMP5 in primary cultures of cardiomyocytes and fibroblasts. In myogenic cells, BMP2 has been shown to up-regulate a series of transcription factors, including GLIS3¹, which usually acts as a negative regulator of transcription and has a specific binding site in the TGF β promoter². This could represent a plausible mechanism by which anti-BMP1.3 antibody could inhibit TGF β expression, meaning that it may block TGF β pathway at both transcriptional and post-translational levels. This concept is now included in the Discussion.

G: Showing shams would be helpful for appreciation of the magnitude of (rescue) effect

Following Reviewer's recommendation, we have now included a sham heart, in which Lox signal is barely detectable in Figure 2h.

Are DPYR and ALD known to be less dependent on BMP1.3/ Lox than other crosslinks?

To the best of our knowledge there are no data to support a reduced dependence on BMP1.3/Lox by DPYR and ALD. For ALD the difference is still significant and for D-PYR the trend is there. Overall, the data indicate less collagen cross-linking and the extent of the differences could be reasonably due to the different sensitivity of the individual assays.

Fig 3

ABC: How to understand that TUNEL is 30 –fold lower than α Actinin? Just a narrower window for detection or another stage of the process, and is this relevant to MOA of (anti-)BMP1.3?

We are not sure to have fully understood this comment. We do not expect all cardiomyocytes being apoptotic and while α -actinin stains cardiomyocyte cytoplasm, TUNEL only stains nuclei, which are relatively small in size compared to total cell area. As also explained below, we used α -actinin as a cardiomyocyte marker and we do not expect variations in its expression upon any treatment. This is now clearly stated in the manuscript.

What is the intervention point of BMP1.3 in apoptosis induction? Is this independent of the apoptosis induction pathway?

As shown in Figure 4, we believe that the anti-apoptotic activity of anti-BMP1.3 Ab is due to its capacity to up-regulate cardioprotective molecules, in particular Bmp2 and 5.

DEF: Why no quantification of α Actinin (cf Fig 4 D)?

As for in vivo experiments, we used α -actinin to label cardiomyocytes and we do not expect variations in its expression. By expressing the TUNEL results as percentage of cardiomyocytes, we do take into consideration α -actinin staining (cardiomyocytes = α -actinin positive cells). We have now clearly specified that α -actinin is only used as a marker of cardiomyocytes.

F: Is this spontaneous apoptosis in culture?

As mentioned in the text, apoptosis was induced by culturing the cells in hypoxic conditions using a dedicated equipment. This information is now also included in the figure legend for clarity.

Fig 4:

b: Why focus on BMP 5? BMP2 is 10x higher (in FB) in CM BMP5 5x higher. Page 13-14 (Discussion) cites some in vitro observations, but what is conceptually more important/interesting? How is expression of different BMPs in vivo?

We agree with the Reviewer that showing the expression of both BMP2 and BMP5 was obvious considering the RT-PCR data. In our original version, we were not able to successfully stain for BMP2 and thus we did not include this data. Urged by this Reviewer's comment, we tried different antibodies and optimized the protocol until we obtained convincing results, which are now included in Figure 4. The other BMPs did not change in expression, consistent with in vitro data.

D: Here no TUNEL? cf Fig 3D-F

Usually, TUNEL is more sensitive than cell number in detecting dying cells. Since in this case we obtained evident results in terms of cell number, we did not feel the need to also show TUNEL staining.

E: siBMP1.3 would have been an interesting too!

As explained above, we silenced BMP1.3 using specific siRNAs, which led to 50% reduction in expression and thus to significant upregulation of BMP2 and 5 but not significant increase in cardiomyocyte survival (see our response to point 1).

F: Living, vs apoptotic in previous

As mentioned before, differences in cell number are more obvious and convincing. In Figure 3 we used TUNEL for consistency with in vivo data (where cell number was not quantifiable). In Figure 4, we preferred to show living

and not dying cell, as we did not have in vivo data to analyze in parallel.

References

1. Bustos-Valenzuela, J. C., Fujita, A., Halcsik, E., Granjeiro, J. M. & Sogayar, M. C. Unveiling novel genes upregulated by both rhBMP2 and rhBMP7 during early osteoblastic transdifferentiation of C2C12 cells. *BMC Res. Notes* **4**, 370 (2011).
2. Dhaouadi, N. *et al.* Computational identification of potential transcriptional regulators of TGF- β 1 in human atherosclerotic arteries. *Genomics* **103**, 357–370 (2014).
3. Xu, J. *et al.* High-Dose TGF- β 1 Impairs Mesenchymal Stem Cell-Mediated Bone Regeneration via Bmp2 Inhibition. *J. Bone Miner. Res.* **35**, 167–180 (2020).
4. Grgurevic, L. *et al.* Circulating bone morphogenetic protein 1-3 isoform increases renal fibrosis. *J. Am. Soc. Nephrol. JASN* **22**, 681–692 (2011).
5. Grgurevic, L. *et al.* Systemic inhibition of BMP1-3 decreases progression of CCl4-induced liver fibrosis in rats. *Growth Factors Chur Switz.* **35**, 201–215 (2017).
6. Zacchigna, S. *et al.* Paracrine effect of regulatory T cells promotes cardiomyocyte proliferation during pregnancy and after myocardial infarction. *Nat. Commun.* **9**, 2432 (2018).
7. Anderton, M. J. *et al.* Induction of heart valve lesions by small-molecule ALK5 inhibitors. *Toxicol. Pathol.* **39**, 916–924 (2011).
8. Mitra, M. S. *et al.* A Potent Pan-TGF β Neutralizing Monoclonal Antibody Elicits Cardiovascular Toxicity in Mice and Cynomolgus Monkeys. *Toxicol. Sci. Off. J. Soc. Toxicol.* **175**, 24–34 (2020).

REVIEWERS' COMMENTS

Reviewer #1 (Remarks to the Author):

The authors have addressed all of my comments.

Reviewer #2 (Remarks to the Author):

The authors have satisfied all of my concerns.

Reviewer #3 (Remarks to the Author):

The authors have greatly increased the clarity of their work, including some points I had apparently misunderstood. The additional analyses provide valuable evidence and support the conclusions with respect to the potential translational potential of anti-BMP1.3. I consider this may now be considered for publication, although still some remarkable observations remain difficult to appreciate. In particular explanation of total shut down of TGF β mRNA by Glis3 upregulation seems a long shot, considering reference 2 listed under the response to reviewers lacks experimental evidence and/but even states "No reports link GLIS3 or the transcriptional repressor ZNF300 to either the TGF- β signaling pathway or atherosclerosis. However, GLI2, a TF very closely related to GLIS3, has been shown to potently induce TGFB1 expression in CD4+ T cells [32].". Might this merit reconsideration?

Sincerely,

Roel Goldschmeding,

Utrecht, The Netherlands

Manuscript NCOMMS-21-00704A

Bone morphogenetic protein 1.3 inhibition decreases scar formation and supports cardiomyocyte survival after myocardial infarction

Response to Reviewers.

We are grateful to Reviewers for their appreciation of our work.

To address the remaining comment by Reviewer 3, we have rephrased the Discussion clarifying the lack of direct evidence showing the capacity of GLIS3 to inhibit TGF β 1 expression and reducing the emphasis on this mechanism.